# The Assessment of the Psychomotor Profile in Children: Preliminary Psychometric Analysis of the Portuguese Version of the Batterie d’Evaluation des Fonctions Neuropsychomotrices de L’enfant (NPmot.pt)

**DOI:** 10.3390/children9081195

**Published:** 2022-08-09

**Authors:** Nídia De Amorim, José Parreiral, Sofia Santos

**Affiliations:** 1POLO UIDEFMH, Faculdade de Motricidade Humana, Universidade de Lisboa, 1499-002 Lisboa, Portugal; 2Centro de Reabilitação de Paralisia Cerebral de Coimbra, 3030-188 Coimbra, Portugal; 3UIDEFMH, Faculdade de Motricidade Humana, Universidade de Lisboa, 1499-002 Lisboa, Portugal

**Keywords:** evaluation, psychomotor development, childhood, psychomotricity, psychomotor profile, validation

## Abstract

Psychomotor assessment is an expanding research field. A deep knowledge of the typical development will allow for better child-centered planning. Due to the scarcity of psychomotor assessment instruments validated in Portugal, our research aims to perform a preliminary analysis of the psychometric properties of the Portuguese version of the Battery for Neuropsychomotor functions evaluation (NPmot.pt). The NPmot.pt was translated and adapted to the Portuguese language/culture and applied to 200 children, 4–12 years old (6 y 10 m ± 0 y 4 m), with (*n* = 150) and without developmental disorders, attending regular school. For content validity, nine experts classified all items according to their relevance, clarity, simplicity and ambiguity. All indexes (IVC > 0.78) pointed out the representativeness of indicators, corroborated by experts proportion agreement (>0.42), and Cohen’s kappa scores (0.02 > k < 0.95). Reliability was confirmed through internal consistency, with Cronbach alphas/split-half (α > 0.45) and temporal reliability (test-retest technique, 0.45 > r < 0.99). Construct validity was analyzed through domains and domains-total correlations, tending to moderate to strong (0.31 > r < 0.92); exploratory factor analysis pointed out an eight-factor solution, explaining 88.5% of the total variance. For discriminant validity, we conducted a comparative study between children with and without developmental disorders that revealed significant differences (*p* < 0.05). The NPmot.pt seems to confirm validity and reliability for Portugal; however, more studies are needed.

## 1. Introduction

Psychomotricity is an area of knowledge that is seeing an increased interest in education and health care in Europe [1], including Portugal [2]. Fundamentally, it involves the interaction between psychomotor development and psychological-emotional functions [3]. Psychomotor Therapy (PMT) is being provided as one of the main supports to children in diverse contexts (schools, hospitals, institutions) and from a three-dimensional perspective: educational, re-educational and therapeutic [2,4], through an evolutionary, developmental and competencial understanding of the process of human growth [5]. PMT, integrating cognition, body movement and emotions, is based on learning-by-movement experiences, on a holistic view of the child, through the unity of body and mind [6]. The goal is to enable the person to act accordingly and autonomously within their own psychosocial context. “The moving body in all its aspects is the cornerstone of the psychomotor approach [distinguishing…] psychomotor therapy from other approaches” [7] (p. 28). PMT is performed by a psychomotor therapist [8].

Recent evidence supports the relevance of the quality of the psychomotor profile in child development [3]. Evidence pointed out the universality (all children go through the same stages) of the specific developmental sequence of different psychomotor skills [9]. The acquisition of motor milestones is an important indicator of developmental trajectory, affecting the overall development of children and their interaction with environmental demands. Further, even though not a formal diagnosis criterion for most developmental disorders, psychomotor limitations are present in several disorder manifestations, such as intellectual and developmental disability (IDD) [10] or autism spectrum disorder (ASD) [11,12], among others. The screening and detection of psychomotor developmental limitations and clinical concerns at various stages of childhood development is essential for the diagnosis, informed planning, progress assessment and monitoring of interventions [13]. The collection of data should be both quantitative (product-oriented) and qualitative (process-oriented) for a more holistic understanding of the movement performance of children. The appreciation of these individual variations (quantitative and qualitative) can provide significant, semiological, and diagnostic indications [14,15,16,17,18,19,20], establishing symptomatic pictures and enabling comparison with peers, aiming to describe in detail the “real value” of the limitation expressed by the child [20]. Current research in the field of PMT warns of the fact that assessment and understanding of clinical pictures of developmental disorders [14,16,17] neglect psychomotor and perceptual-motor limitations.

The lower quality of psychomotor performance of children with developmental disorders results from the complexity of the interaction of the cognitive-motor and sensory systems [21,22], with a less positive impact on the relationship with involvement. Changes in psychomotor skills that involve tonus and balance [23,24], with repercussions on gait [25], strength and coordination of the upper and lower limbs are reported in children with developmental disorders [10], interfering with performance in other tasks involving, among other skills, eye-hand coordination [26]. Moreover, the greater the cognitive impairment, the greater the difficulties in terms of activities with the ball, balance and manual dexterity [27]. The balance and integration of sensory stimuli’s limitations are also observable in children from six to ten years. In their comparative studies between children with and without developmental disorders, Shum and Pang [28] and Geuze [29] noticed significant difficulties in balance and integration of sensory stimuli. Children with T21 tend to use their left hand, which increases with age vs. typical peers that tend to use their right hand [30]. Manual proficiency was clearly slower in the T21 group, regardless of age, although it improved with increasing age. The right hand was the most proficient in both groups. Children with developmental disorders present significant limitations on coordination skills: children with ASD demonstrate a more generalized commitment concerning gestural performance and tend to present a higher number of errors/distortion in imitated gestures than the other groups [31]. Children with ADHD tend to fail tests that assess body image, with a high level of impulsivity index [20]. In terms of auditive attention, Vaivre-Douret [20] evaluated 20 children with ADHD (four from each age group) and found few correct choices with a high level of impulsivity index. Miyahara and collaborators [32] compared the fine motor skills of children with Asperger’s Syndrome (SA) and Specific Learning Difficulties (SLD), noting that the former demonstrates better manual dexterity and lesser skills in handling balls.

Despite the evidence, research does not enable us to understand the type and nature of such disturbances. Few instruments rely on a developmental approach. The relation between neurodevelopmental processes and central nervous system maturation needs to be more explored for a clear understanding of the origin of such limitations [12] and to act accordingly. Despite this, there are some motor instruments (Lincoln–Oseretsky test [33], Motor Coordination Scale [34], and the Child Movement Assessment Battery M-ABC [35]) that are not developmental [12] and show higher levels of complexity in association with increasing age [20]. Most are dependent on learning and practice and do not allow drawing a minor deficits symptoms profile (neurological soft signs) during an overall motor skills evaluation [12]. In addition, these assessments measure mostly competencies related to the practice/training of a task [20]. Some traditional psychomotor tests involve the assessment of right–left discrimination and spatial orientation as an implicit coordinate system [9]. Body awareness/image (recognition of body parts, reproduction of movements, drawing figures) is another targeted dimension of other psychomotor tests, particularly in adolescence [36], but lacking in childhood. There are a few tests that do not fully satisfy the metric requirement; when analyzed with little psychometric data on the validity and reliability of models and tests, most do not have cultural adaptation, which limits their use and does not allow comparisons. Some findings in reliability are not acceptable [37], others present low correlations between domains in children [38,39,40], and the confirmatory factorial analysis does not validate the original factor model [41].

Psychomotor development is characterized by periods of growth and abilities acquisition (2 to 5 years) followed by stabilization moments for the specialization of those skills (>6 years [20]). The typical development respects the natural sequence of the central nervous system maturation and all the subsystems associated that contributes to a harmonious development of psychomotor skills [6]. The normative evolution allows placing the child in its physiological (genetic) maturation and detecting a functional disorder with a symptomatic value of a neurological, psychomotor or psychological origin. There is a need to establish and analyze the psychomotor profile of children [42]. This assessment should integrate all dimensions of childhood and not be focused solely on behavior analysis [17,43]. Psychometric assessment provides data about performance in several activities on a norm-referenced comparison [44]. However, for the psychomotor therapist, this is a very restricted vision of the child [6,17,43,45] because the performance should not be reduced to a single number to be compared with a pre-established norm. That performance should be contextualized, and qualitative observation (precision, quality) should, therefore, be considered an important key factor in the assessment [20]. One of the major contributions of PMT assessment is not just to be focused on the final motor performance (product), but also on how the child integrates, programs, executes and plans their thoughts vs. observable behavior [6]. However, there are no updated norms about children’s psychomotor development, and most of the tests used are more focused on cognitive and verbal competencies rather than psychomotor ones [46].

The most consensual model of the human psychomotor system [6,20,47] comprises a set of psychomotor factors, interdependent and in constant interaction, that differs in number and naming, with weak to strong correlations between them. The Psychomotor Battery developed by Vítor da Fonseca [42] was based on an extensive literature review (theoretical approach) but without empirical data. Correlations between factors ranged from 0 (lateralization and fine motor skills) to 1 (fine motor skills and space-time, body awareness and global motor skills). No exploratory or confirmatory analyses were performed. In France, the Vaivre-Douret [20] model comprises nine functions: tonus, gross motor skills, laterality (tonic, spontaneous/gestural, psychosocial and usual), manual praxis, tactile gnosis, hand-eye coordination, spatial orientation, rhythm and auditive attention. 

Based on a selection of (psychomotor) items from other tests to evaluate psychomotor development, in 2006, Vaivre-Douret proposed the creation of the Neuropsychomotor Function Evaluation Battery-NP-MOT [20], updating the application rules in order to respond to the psychomotor therapist’s difficulty in assessing psychomotor development [17,19,20,43]. The NP-MOT covers the motor components and enables the assessment of neuromotor functions: extrapyramidal (related to general and daily activity involving both involuntary and voluntary movements, mastery/specialization, and speed [6,42], at rest, passive, static, automated and involuntary and voluntary movements), cerebellar (e.g., the ability to remain static and dynamic balance with the collaboration between involuntary and voluntary movements) and pyramidal (reflected at the voluntary movements). The NP-MOT assesses from two different, but essential, points of view—the mastery and specialization (quality) of the movement and the speed (quantitative aspect) of the involuntary and voluntary movements produced. The NP-MOT may be applied for diagnosis, identification and description of strengths and areas to be promoted, contributing to a better semiology and a more personalized and effective intervention approach [18,19,43,47]. This new battery was developed to monitor physiological evolution, identify significant deviations from typical development and detect minor functional disorders that might have a pathologic impact on learning processes [12,20]. The establishment of the psychomotor profile is an emergent need for psychomotor therapists to perform a comprehensive, accurate and reliable measurement considering all these factors [38].

There is a call for appropriate testing to identify psychomotor skills across countries [39,40], not only for cross-cultural comparisons but also to develop a common language among psychomotor therapists [38,40]. In France, as well as other countries (including Portugal), there are no up-to-date standards for child neurological development due to the absence of a valid and reliable instrument to assess children’s psychomotor functions with quantitative evidence to establish the psychomotor profile and the respective cut-off scores. Further, it also appears that the tests used to assess evolutionary development and maturation are mostly oriented toward the assessment of cognitive and verbal, excluding, directly or indirectly, the assessment of psychomotor skills [46,48]. Moreover, the quantitative assessment of the child is not enough for the psychomotor therapist, who needs to also gather data about behavior analysis (during assessment). According to Fonseca [42], the interpretation of data from a psychomotor assessment should complement the performance (product) with data about how the child integrates, programs, executes and regulates their behavior (process).

It is an emergent standard instrument to assess psychomotor functions and analyze the integration of sensory and neuromotor systems linked to the central nervous system’s maturation [20]. Most of the psychomotor assessment instruments are outdated, some are based on classical and old statistical analyses and others do not meet the quality metrics that a diagnosis instrument should have [13]. In the national scenario, despite the availability of some psychomotor tests for adults and elderly [38,49,50], little research has been focused on the analysis of metric qualities, such as validity and reliability, especially with children. The use of instruments with robust psychometric properties will allow the gathering of data-based evidence, fundament to the PMT conceptual model, for a more valid child-centered plan and to assess the psychomotor interventions’ effectiveness [9,13,40].

Therefore, and because of sparse psychometric research, our goal aims to present studies in the field of validation of the Portuguese version of NP-MOT for children between 4 and 12 years, with and without developmental disorders, to contribute to a valid measurement instrument for establishing the psychomotor profile, through a preliminary analysis of the psychometric properties: content validity, reliability, discriminant and construct validity.

The French battery was chosen for several reasons. This instrument is the result of ten years of research in the field of French neurology and is an innovative psychomotor assessment tool that promotes a neuro-functional understanding of the child’s brain organization. This feature allows us to detect and analyze the nature of various developmental changes [51]. The NP-MOT is a clinical, standardized, valid and reliable instrument [20] that aims “to capture maturational aspects in the ability to perform and quality of performance, reflecting the development of neurological mechanisms contributing to movement timing, motor control, motor coordination and motor execution” ([48], p. 4). It is also a developmental and age/maturation-related assessment, with identical subtests (qualitative and quantitative developmental norms) across ages and without a global score from the battery. Further, it comprises a quantitative and qualitative approach to the assessment of neuromotor functions, which allows detecting neurological soft signs often never systematically addressed [48]. The mere translation of tests is not enough to be applied within different age or sociocultural subgroups, and all the procedures suggested in the literature [52] were followed. There is still a lack of knowledge and absence of standardized instruments of body structures and functions in Portugal. “It provides information on maturation levels for each of the functions explored, independently of other functions [entailing] one or more tests, some of which consist of various items” ([48], p. 2).

## 2. Materials and Methods

### 2.1. Participants

Our convenience sample involved 200 children between 4 years and 12 years and 6 months old (6 y 10 m ± 0 y 4 m), with (*n*_female_ = 19 and *n*_male_ = 31) and without developmental disorders (*n*_female_ = 86 and *n*_male_ = 64), attending regular and public schools, enrolled from preschool education (*n* = 71) to the second cycle of basic education (*n* = 129), in the district of Beja. As inclusion criteria were: the ages of participants (4 > x < 12 years), meeting the original standardization sample, but extending to 12 years because the PMT is one of the supports provided at these ages in the school community; attendance at a regular school; and, in case of children with a developmental disorder, existence of prior clinical identification of developmental disorder (ASD, IDD) in their educational process. Within the scope of the exclusion criteria were defined: age (x < 4 e x > 12) and the existence of an associated sensory and/or motor disability that made the application of tasks impossible.

The test–retest was analyzed through the application of the test twice, with an interval of 2–3 weeks, to 50 children (males *n* = 25) and five children at each age stage (6 y 1 m ± 0 y 4 m) who respected all inclusion criteria of the study.

The sample was divided into six age groups, according the following distribution: group 1–4 years to 4 years and 9 months (*n*_female_ = 25 and *n*_male_ = 4); group 2–4 years and 10 months to 5 years and 8 months (*n*_female_ = 13 and *n*_male_ = 28); group 3–5 years and 9 months to 6 years and 6 months (*n*_female_ = 7 and *n*_male_ = 0); group 4–6 years and 7 months to 7 years and 5 months (*n*_female_ = 20 and *n*_male_ = 8); group 5–7 years and 6 months to 8 years and 5 months (*n*_female_ = 30 and *n*_male_ = 22); and group 6–8 years and 6 months to 12 years (*n*_female_ = 10 and *n*_male_ = 13).

### 2.2. Instrument

The Portuguese version of the NP-MOT (NPmot.pt) was translated and adapted from the standard French version [20]. The NPmot.pt aims to assess the neuropsychomotor skills of children between 4 and 12 years, to understand the normative development of neuromotor, neurosensory and perceptual integration functions, as well as to establish and understand the neuropsychomotor profile of children (strengths and weaknesses) with and without typical development. The data obtained will contribute to the planning and implementation of effective interventions.

NPmot.pt, in its Portuguese version, maintained the same original structure, with all 52 items distributed over 9 psychomotor domains [20]: Tonus (residual, action, support, attitude tonus and rotellian reflexes); Gross Motor Skills (dynamic and static balance, gait); Laterality (spontaneous, daily and psychosocial gestures), Manual Praxis (repetitive and alternating movements), Tactile Gnosis, Hand-Eye Coordination, Spatial Orientation (in themself, in the other, about an object and through a map), Rhythm (time, auditive-kinesthetic, auditive-perceptual-motor) and Auditive Attention (selected and sustained). The domains can be applied isolated (if there is a need to understand only one specific function), but the global assessment of all domains will allow a better definition of a global neuropsychomotor development profile, contributing to the definition and analysis of differential diagnoses [20].

The items are rated by assigning two grades (one quantitative and another qualitative). The quantitative refers to the child’s performance and varies from 0 (does not perform) to 2 or 5 (corresponding to the best performance, depending on the items). This assessment is conjugated by a qualitative observation of the components of the gesture (quality, precision, speed). The qualitative observation is characterized by the performance of a harmonious movement or with visible changes (resistance, tremors, anxiety). The test should be norm-referenced, and data should be interpreted according to a comparison with the standard values [20]. It will be possible to define a global neuropsychomotor profile, allowing the identification of strong areas and areas to promote a more adjusted and personalized neuropsychomotor intervention [20].

The content validity of the original version was obtained by expert agreement (76% to and 98%). Reliability was analyzed by temporal stability (test-retest technique), in which scores ranged from 0.70 (Tonus) to 0.96 (Laterality), and internal consistency through Cronbach’s alpha showed scores above 0.70 [20,53,54]. The sensitivity of the test was assessed through statistical analysis, showing a significant increase in values from age to age. For the criterion’s validity, the author analyzed the correlations between the NP-MOT and the Lincoln–Bruininsky–Oseretsky Test of Motor Proficiency Psychomotor Development Battery [33]. Both scales were applied to 50 children with and without developmental coordination disorder (DCD), and IQ values above average. Correlations were strong (0.72 > r < 0.84), and the comparative study indicated that children with developmental coordination disorder had lower results (1 or 2 standard deviations) in almost all domains.

### 2.3. Procedures

#### 2.3.1. Translation and Adaptation

The present investigation respected all ethical procedures inherent to the validation process of an evaluation instrument [52]. Firstly, it was obtained the original author’s permission to carry out the adaptation. After translating and adapting the procedures to guarantee that the construct was understood in the same way across language and cultural groups, content validity was performed to confirm equivalence between both versions. Particular attention was given to context–cultural–linguistic adaptation (experts agreement), trying to use natural, simple and clear, not ambiguous language. The initial translation was focused on functional rather than literal equivalence. Therefore, two Portuguese and French-native speakers, both experts in PMT, respectively, performed a forward and back translation. The two versions were compared by two practitioners to clarify instructions and response options and to sustain a similar structure through all tasks. All disagreements were solved by the third investigator. A pre-final of the test was established. Then, the Portuguese version, with a clarification of the goal and target group of the test, was sent to 9 experts, selected based on their academic, methodological and professional experience in PMT and children development, to classify and rate each item according to its relevance, clarity, simplicity and ambiguity. However, because a theoretical-qualitative approach is not enough to guarantee the suitability of adaptation and the scales’ equivalence, a field test was performed.

#### 2.3.2. Administration and Statistical Analysis

After obtaining Ethical Approval for the study, contacts were carried out with several schools to request authorization to apply the NPmot.pt to students between 4 and 12 years, with and without developmental disorders. For this matter, informed consent with the explanation of research (goal, procedures, ethical topics, confidentiality, anonymity, analysis and contributions) was sent to all School Board Directors. In case of accordance, a similar document was sent to primary caregivers. All participants (with and without developmental disorders) signed the informed consent/gave oral consent. The application of the NPmot.pt was conducted according to its protocol, in scholar gyms, according to children’s schedules and availability, to minimize the interference with other curricular and extracurricular activities. Each application took about 90 minutes on average, depending on the child’s characteristics. There were no missing data, and all data collected were analyzed. 

We investigated three types of validity. For the content validity analysis, two distinct phases were carried out [55,56,57]: a developmental descriptive based on an extensive literature review about the construct and assessment tools, and an empirical one that required the participation of 9 experts (parents, practitioners and academics). Of those experts, four have a Ph.D. in Special Education and two in Human Kinetics (PMT), three are experts in validation methodology, two have a master’s degree in PMT, and three are psychomotor therapists that work with children with and without developmental disorders in schools, for the last ten years, in Portugal. Each expert rated all items based on relevance, clarity, simplicity and ambiguity through a four-point Likert scale ranging from 1 (much irrelevant) to 4 (much relevant). Then, these four options were transformed into a dichotomous scale with items rated previously with 1 and 2, considered not having content validity, and 3 and 4 as valid [55,56,58]. All content validity indexes (CVI) were determined: for items, scale average (CVI_A), and universal agreement (CVI_UA). However, this analysis is insufficient because these indexes do not provide data about the adjustment of agreement expected by chance [55,58]. Therefore, the proportion between experts and Cohen’s kappa was also performed. The cut-off scores assumed for each index were [55]: CVI > 0.78, CVI_UA > 0.80, CVI_A > 0.90. Cohen’s kappa interpretation was based on the following values: poor (k < 0.40), moderate (0.41 > k < 0.60), substantial (0.61 > k < 0.80) and excellent agreement if k > 0.81 [58,59]. 

For reliability, we employed Cronbach’s alpha and split-half to analyze internal consistency, assuming that scores below 0.70 are inadequate, between 0.71 and 0.80 are acceptable, exceptional above 0.81 [60] and preferably above 0.90 for diagnosis purposes [60]; and the correlations coefficients of Pearson (technique test-retest: two applications of the test within a 2–3 week interval) to examine temporal stability: weak if r < 0.40, moderate when 0.41 > k < 0.69, high between 0.70 and 0.89 and very high if superior to 0.90 [61]. The construct analysis considered domains-domains and domains-total, and an exploratory factorial analysis (EFA) to explore the underlying structure of the construct was measured. Finally, the discriminant validity to evaluate differences between children with and without developmental disorders was analyzed through the parametric techniques of t-student and One-way ANOVA, using a 95% significance level. The choice of tests for the statistical analysis of the data collected was based on the fulfillment of the assumptions necessary for parametric tests. In this sense, the dependent variables presented a normal distribution in both groups, most of the cases being close to the mean value. The existence of homogeneity of variances (homoscedasticity) was also verified; that is, the variances of the dependent variables were homogeneous in all groups. The test used to verify Normality was the Kolmogorov–Smirnov, and the Levene test was used to test the homogeneity of variances. The sample size (*n* = 200) also consolidated the decision to use parametric tests. 

The application of the battery is simple, individual and uses little material; the original protocol was followed by all examiners, starting with physical observations, as suggested [20]. All examiners had access to a suitcase with the materials.

Data processing was performed using Microsoft Excel version 14 (14.0.7268.5000), 64-bit, developed by Microsoft, Lisbon, Portugal and Statistical Package of Social Sciences (SPSS) software, version 25, developed by IBM, Lisbon, Portugal.

## 3. Results

Content validity measures the degree to which an instrument covers the construct that we want to measure [56]. After the translation, a reconciliation procedure aimed to guarantee the equivalence of the Portuguese version, and some clarifications were made. Due to the limitations of such a qualitative analysis, an empirical approach was also performed through expertise agreement [55,56,58]. This method required multiple levels of the agreement by a nine-expert committee. All items (52) were considered relevant, and only the items that do not account for at least 0.78 [55] in any of the other categories are presented (Table 1).

Because some scores in clarity, simplicity and ambiguity were lower than 0.75, particular attention was given to the wording and readability of items. Reformulations were performed in nine items (e.g., a simple explanation was introduced about a specific item because only the name was not understandable to the experts) as well as a new order in presenting the items for a more effective application (e.g., applying the Tonus exam last, applying the items of Bi-manual and Digital Praxis next to the syncinesies exam or starting the evaluation beginning in the Registration Book 2). Item names were simplified and clarified (e.g., item 14: Lateralization was replaced by Tonic-Manual Lateralization and Manual Daily Lateralization), and a small description was added for faster identification of what was measured. Despite the lower scores in clarity, simplicity and ambiguity, the relevance category presented indexes above 0.78. The CVI_AU is related to the proportion of items considered relevant by all experts, ranges from 0 (total disagreement) to 1 (total agreement) and should be superior to 0.80. The CVI_A is the ratio of all relevant options (3 and 4), in each of the categories, by the total number of experts and should be above 0.90 [55]. Both indexes were significant for the relevance level.The decision was to keep all items. Due to CVI limitations, the agreement between experts was also calculated (Table 2), and there was an excellent agreement among all experts concerning the relevance of items. 

Only in clarity (0.41 to 0.67) and simplicity (0.58 to 0.77), the indexes were below 0.80, but the agreement was still considered moderate to substantial. The number of experts may influence these results. Many of the items were reformulated, and these initial analyses played a critical role in the adaptation-formulation, understanding and readability of questions. Finally, Cohen’s kappa (Table 3), to analyze the agreement level between each pair of experts [62], was calculated. Considering the number of experts (*n* = 9) the agreement tended to be positive, ranging from moderate (>0.40) to excellent (0.96), although some lower scores related to experts 5 and 7 (k = 0.05), 7 and 9 (k = 0.02) and 9 and 3 and 4 (k = 0) [58,59]. The lowest scores may be explained by the absence of disagreement between the pair of experts, which may lead to a decrease in these experts’ global agreement [55].

After confirming the content validity, the final version of NPmot.pt was established, and a test pilot was conducted. The battery was then applied twice within 2–4 weeks to 50 children to analyze temporal stability [54,61]. Further, reliability was also analyzed in terms of internal consistency through the calculation of Cronbach’s alpha, Split-Half and intra-class correlation coefficient (ICC) (Table 4). The reliability indexes ranged from 0.45 to 1, indicating NPmot.pt as a reliable instrument, although there was a lower score on digital gnosis (0.45).

The construct validity was analyzed through domain inter-correlations (Table 5) using Pearson’s correlation coefficient [53,63] to measure the degree of independence of the domains among themselves [54]. Domains correlated with each other moderately (0.30 > x < 0.60) [57] and at strong levels, ranging from 0.31 (Laterality and Tonus) to 0.92 (Spatial Orientation and Gross Motor Skills domains). Only the score of 0.27 between the Tactile Gnosis and Tonus stood out as weak correlation domains.

EFA (Table 6) was also performed to explore the organization of items by domains [20] using the maximum likelihood estimation method, the Kaiser–Meyer–Olkin index (KMO) and the Bartlett sphericity test [54]. The factor loadings pointed out the multidimensionality of the construct on an eight-domain structure (and not nine as in the original version). The first two components explained about 69.2% of the overall variance. Eigenvalues greater than 1, a minimum of 5%, explained variance per component of the scree plot; factor loadings of 0.40 and above were the criteria assumed for the extraction [54].

One of the important characteristics of instruments of this nature is the discriminant validity for distinguishing profiles between groups [20]. We analyzed several variables: gender, diagnosis (with and without developmental disorders) (Table 7) and age (Table 8 and Table 9). Parametric Student’s t-tests were used for dichotomous variables (i.e., gender and diagnosis) and One-way ANOVA with post-hoc Scheffé, given the uneven number of participants per step age.

Statistically significant differences were observed using the Student’s t-test at the level of neuropsychomotor skills of participants with and without developmental disorders in all domains (*p* < 0.01), as expected. In terms of gender, there were only differences found in Laterality and Rhythm. 

The results show that the mean scores in all domains were similar, despite the increase in age. This corroborates the conceptual model behind the construction of this instrument since the author mentions that the same items are applied to all children, with standardized classes that allow framing the child in its age group and not with all children aged between 4 and 12 years. Lower values were also observed in the older age group, which may be contradictory, but the fact that in this age group, there is a high percentage of children with developmental disorders may justify these results.

Overall, there were statistically significant differences between the children between 4 and 5, 8 years old and the older ones. Further, group 6, which included the oldest children, also tends to present significant differences from the others, maybe because children with developmental disorders are included in this group. These findings seem to be aligned with the developmental approach of the instrument NPmot.pt. The critical period of acquisition of neuropsychomotor skills seems to be the range of 4–6 years, showing a stabilization and maturation from 8 years.

## 4. Discussion

This research analyzes the metric properties of a neuropsychomotor assessment tool for children aged 4 to 12 years. Validity and reliability are the prerequisites to guarantee the quality of any instrument, and because they are not “transferable” from the cultural/age group, they should always be analyzed with a new distinctive group [39]. According to the authors: “reporting of reliability and validity properties of results from motor competence assessments should be a common occurrence in peer-reviewed journal articles” (p. 1778). The starting point was the cross-cultural adaptation because the mere translation of a test is not enough and should be avoided due to the lack of rigorousness in data interpretation [38,50,52,64]. Special attention was paid to the translation and adaptation of the battery [52] to the Portuguese language and cultural values. This adaptation required an exhaustive literature review of the construct and its evidence-based indicators. The translation process considered the inclusion of the expert’s native speaker of both the original (French) and Portuguese versions for an initial content and semantic equivalence. After, it was asked to three psychomotor therapists to analyze the battery in order to adequate the wording and instructions of items. Finally, the empirical procedure involved nine experts, whose judgment agreement allowed establishing a final evidence-based version of the correspondence between items and respective domains. In the case of having more than six experts, CVI should be superior to 0.78 [52,56]. 

In our findings in the Relevance category, only four items (items 13—Rotellian Reflexes, 32—Psychosocial Laterality, 33—Symmetrical bi-manual Pronation-Supination, and 34—Simultaneous Asymmetrical Bi-manual Pronation-Supination) presented a 0.78 score, maybe due to the existence solely of the name and not what was asked in each task. Based on experts’ judgments and qualitative comments, language was clarified and simplified. Further, some of these items seemed redundant for some experts (qualitative comment), and therefore, special attention was given to them. All the rest of the items scored higher (0.89) in their representativeness as an indicator of each domain. The decision was to keep all items, although the lower scores in other categories (clarity, simplicity and ambiguity) led to some reformulations and clarifications, to make the items more understandable, avoiding the use of ambiguous terms. One of the main changes was to make clearer the naming of some items (e.g., item 13—rotellian reflexes) as well as to simplify/add explanatory instructions of its tasks. Some other items (5, 7, 11, 12, 13, 14, 15, 28, 29, 32, 49, 51 and 52) needed a reformulation on semantic, grammar and sentence construction (e.g., item 5, in the French version: Extensibilité des angles poplités, to the Portuguese version: Extensibilidade do poplíteo (popliteal extensibility)). According to the perspective of the original author, the tonus domain (items 5 to 13) needed more specific items rather than using more comprehensive items as other scales (M-ABC or BOPMT-2). Vaivre-Douret [20] considered this domain as neglected, despite its important role in gathering information about child development. The CVI of each item was corroborated by the AU/IVC-E (0.83) and the M/IVC-E (0.97). Experts assumed the relevance of all items of NPmot.pt. There are no data about the psychomotor battery used in our country.

This accordance was corroborated in the proportion of agreement among experts with scores higher than 0.73 (in the ambiguity category). Again, when it comes to relevance, the experts tend to present an excellent agreement (>0.96), very similar to the original version [20]. The poor agreement scores tend to be associated with experts 5, 7 and 9 that presented, however, an exceptional agreement between experts in relevance (0.98 to 1), and moderate to strong on clarity (>0.42), simplicity (>0.80) and ambiguity (>0.58). The lowest scores may be explained by the reduced disagreement between the pair of experts, which may lead to a decrease in these experts’ global agreement [55]. Further, Cohen’s kappa is sensitive to the types of disagreement [59], which may influence the final scores. Although our findings seem to be in line with the original study [20]—which also presented a very good to excellent agreement proportion (0.76 to 0.98), there are no further details about the selection of items and empirical description of the content validity indexes. Therefore, we were unable to compare our findings with the original ones. However, both scales are based on a similar multidimensional model. Then, the reliability of the NPmot.pt was analyzed. Our findings suggest that NPmot.pt has overall good to excellent reliability, although further studies should be conducted with larger samples. Internal consistency presented excellent scores (α > 0.85) in all domains, except in *hand-eye coordination* (0.79), which was acceptable, and *digital gnosis* (α = 0.45). The internal consistency of the total scale was 0.86. These findings, corroborated by other reliability indexes calculated, seem to be aligned with Nunnaly’s [60] reminder of the need to have scores above 0.90 for diagnosis purposes. All indices point out the battery as a reliable instrument, corroborating the original scores [20], although it needs closer attention, especially in the *Tactile Gnosis* domain, which in France reported an acceptable 0.83 [17,20]. Still, more studies are recommended. 

There were consistently moderate to high correlations between NPmot.pt domains, varying from 0.31 and 0.92, except tonus and digital gnosis (0.27), with a weak correlation. This may be explained by the need for a certain level of attention and proprioceptive integration that is not dependent on the tonus muscle. Another hypothesis is that these scores may reflect the non-maturation of these skills, particularly in children with developmental disorders [20]. This score should be further analyzed since it is the domain that is consistently presenting lower scores. The domain-to-domain correlations show moderate to strong correlation, which is aligned with the original version [20] and higher than those reported by Fonseca [6], and seems to support construct validity. Each domain of NPmot.pt has different indicators, and domains are related to each other. The EFA produced consistent findings with previous studies [20] and conceptualizations [6] of a multidimensional construct, although in our sample, the indicators seem to be organized into eight domains, explaining 88.5% of the total variance. 

Finally, in an attempt to understand the existence (or not) of differences in terms of psychomotor skills in a set of variables (age, gender and diagnosis, i.e., with and without the developmental disorder), a comparative study of the psychomotor skills of the participants was carried out. There were significant differences between males and females, which were not found in the original study [20] in the domains of Laterality and Rhythm, with a tendency for slightly higher mean values in females. This idea was already reported by Eckert [64], who mentions the precocity of female motor development, except in the domains of Tonus and Gross Motor Skills, where higher values are expected for males. However, it should be noted that the majority of the participants with a developmental disorder were male.

Analyzing mean scores confirms group differentiation between children with and without developmental disorders in the neuropsychomotor profile, demonstrating potential clinical significance, i.e., NPmot.pt appears to be able to differentiate children with typical from atypical development. Children with typical development score higher than children with developmental disorders. This finding seems to be aligned with others [11,12,13,21,23,24,25,26,32,33,38], and in direct association with cognitive, executive and motor limitations [12,65,66,67], being of particular interest the muscle tone alterations [24], balance and gait [23,25], eye-hand coordination [26,68], laterality (preference and proficiency) and space-time [30]. As expected, significant differences were found between all groups, especially group 6, which despite having older participants, also included children with developmental disorders. Findings seem to corroborate the developmental trajectory until approximately 8 years old [20,64], and by that time, there is a visible tendency for a period of stabilization and consolidation of acquired skills [20,64].

However, findings should be interpreted with caution due to some limitations. The experts’ committee should have involved parents and teachers of children with and without developmental disorders for adding other perspectives on the importance of each item in daily life. The sample was reduced and geographically circumscribed, and there was no control for the level of severity of developmental disorders. The recommendation is to continue to apply the NPmot.pt to a representative, significant and stratified sample to deepen the hierarchical structure of the test and establish the cut-off scores for a more precise and rigorous diagnosis of developmental delays. The establishment of these cut-offs assumes an essential role in order to avoid incorrect/false diagnoses with adverse consequences in children’s lives. The criterion validity is another step that should be performed. The comparison between groups (ages, diagnosis, level of severity, etc.) is also a suggestion for future research with practical implications. The analysis of how psychomotor functions correlate with other variables (e.g., intellectual functioning, adaptive behavior, scholar performance) is also another topic to be explored.

In this particular research, our results will allow an evidence-based use of the psychomotor battery for establishing the psychomotor profile of children between 4 and 12 years. This is a pioneer study due to the scarcity of standardized neuropsychomotor assessment instruments in the context of PMT and the urgent need for a valid test to be used with children. The development of a theory and the design of a sound psychometric measure to gather valid data is essential for several purposes, both in clinical practice and in research [69]. If instruments are not valid or reliable, results obtained cannot be used with confidence, compromising the effectiveness of neuropsychomotor interventions. This data will allow gathering data-based evidence to be used to test the theory/conceptual model of PMT, which is it being supported by the data provided from that measure, and may be used in decision making [13], such as a more valid children-centered plan [2]. It will also allow establishing the neuropsychomotor profile and setting the cut-off scores for determining significant limitations (above average) in neuropsychomotor skills, which will positively impact intervention [6]. Moreover, it will also contribute to the assessment of the neuropsychomotor interventions [9,13,40]. The existence of such an instrument will also contribute to strengthening the psychomotor therapist’s (national and international) identity and the relevance of PMT application to several vulnerable groups [2,13].

## 5. Conclusions

This research aimed to present the cross-cultural adaptation of one of the most recent instruments in the PMT field, as well as the preliminary analysis of reliability and validity qualities. The NPmot.pt seems to be a reliable and valid instrument and could be used for assessing and screening the neuropsychomotor profile of children with and without developmental disorders, appearing to be useful in educational and therapeutic contexts, providing semiological and diagnostic indications, and giving guidance for the development of more precise children-based planning, within the PMT field. Our study goes beyond the mere translation or the proportion of experts’ agreement. The literature review allowed a major understanding of the construct that is intended to be measured that should be understood in the same way across language and cultural groups [52], being the foundation of valid cross-cultural comparisons [40,52]. Judgments of the construct-item match and suitability for the language groups involved were one of the main concerns, and the inclusion of an expert committee was essential for content validity analysis, also introducing empirical data. The equivalence of the structure of the test was guaranteed, and all items were considered representative. The battery is also reliable. However, particular attention should be given to Tactile Gnosis. Moderate to strong correlations and the potential organization of all indicators by eight domains seems to corroborate the multidimensional conceptual model.

The NPmot.pt distinguishes children with typical developmental from those with developmental disorders and points out the developmental trajectory during childhood. The Portuguese version comprises a quantitative record of the child’s performance and incorporates a qualitative holistic approach, aiming to screen minor cerebral disorders that might not be captured through a global performance assessment [70], but with an impact on learning functions (talking, writing, etc.). In the field of PMT, there is a need to have a gold standard that is consistently used across countries and age groups. It is equally important, to examine neuropsychomotor functions in a developmental approach, which promotes cross-cultural studies and group comparisons. Besides the individual profile, the use of valid data collected by the NPmot.pt will assist the assessment of the interventions’ effectiveness—through a comparison of scores over time and a more common language among PMT therapists across the world.

## Figures and Tables

**Table 1 children-09-01195-t001:** Content Validity Indexes (CVI) of NPmot.pt (items and scale).

	Relevance	Clarity	Simplicity	Ambiguity
Item 1	0.89	0.67	1.00	0.56
Item 5	0.89	0.44	0.67	0.56
Item 6	0.89	0.67	0.89	0.67
Item 7	0.89	0.22	0.56	0.33
Item 9	0.89	0.78	0.89	0.89
Item 13	0.78	0.67	0.56	0.44
Item 32	0.78	0.44	0.22	0.44
Item 33	0.78	0.67	0.78	0.78
Item 34	0.78	0.78	0.78	0.78
AU/IVC-E	0.83	0.10	0.37	0.29
M/IVC-E	0.97	0.78	0.87	0.82

**Table 2 children-09-01195-t002:** Proportion of agreement among experts.

	Relevance	Clarity	Simplicity	Ambiguity
E1	1	0.81	1	0.80
E2	0.96	1	0.98	1
E3	0.96	0.94	0.94	0.77
E4	1	0.92	0.98	0.98
E5	0.98	0.42	0.80	0.58
E6	0.98	0.80	0.86	0.80
E7	0.98	0.67	0.88	0.73
E8	1	0.63	0.81	0.81
E9	1	1	1	1

**Table 3 children-09-01195-t003:** Content validity—Cohen’s kappa.

Experts	1	2	3	4	5	6	7	8	9
1	1	0.21	0.56	0.25	0.77	0.62	0.63	0.67	0.73
2		1	0.92	0.79	0.76	0.94	0.95	0.95	0.96
3			1	0.30	0.77	0.67	0.69	0.62	0
4				1	0.93	0.87	0.90	0.90	0
5					1	1	0.05	0.21	0.21
6						1	0.21	0.54	0.57
7							1	0.10	0.02
8								1	0.45
9									1

**Table 4 children-09-01195-t004:** NPmot.pt reliability indexes.

Domains	Test- Retest (*r*) (*n* = 50)	Internal Consistency *α* (*n* = 200)	Split-Half (*n* = 200)	Correlation Coefficient Spearman-Brown (*n* = 200)	Correlation Coefficient Guttman (*n* = 200)
Tonus	0.85	0.87	00.88	0.99	0.92
Gross Motor Skills	0.90	0.90	0.90	0.95	0.86
Laterality	1	1	1	1	1
Manual l Praxis	0.93	0.93	0.93	0.86	0.79
Hand-eye coordination	0.79	0.79	0.79	0.81	0.79
Tactile Gnosis	0.45	0.45	0.45	0.46	0.45
Spatial Orientation	0.91	0.91	0.91	0.91	0.89
Rhythm	0.92	0.92	0.92	0.92	0.92
Auditive Attention	1	1	1	1	1
Total	0.86	0.86	0.86	0.88	0.85

**Table 5 children-09-01195-t005:** NPmot.pt domains’s intercorrelations (*n* = 200).

Domains	T	GP	L	MP	TG	HEC	SO	RI	AA
	Pearson’s Coefficients Correlations								
Tonus	1								
Gross Motor Skills	0.48	1							
Laterality	0.31	0.54	1						
Manual Praxis	0.43	0.89	0.41	1					
Tactile Gnosis	0.27	0.63	0.53	0.54	1				
Hand-eye coordination	0.31	0.68	0.38	0.63	0.47	1			
Spatial Orientation	0.39	0.92	0.44	0.89	0.60	0.60	1		
Rhythm	0.57	0.91	0.57	0.80	0.66	0.69	0.80	1	
Auditive Attention	0.48	0.77	0.47	0.65	0.56	0.59	0.70	0.80	1
Total	0.47	0.76	0.52	0.69	0.58	0.59	0.70	0.76	0.67

*p* < 0.05.

**Table 6 children-09-01195-t006:** NPmot.pt exploratory factorial analysis (structure matrix; *n* = 200).

Component Matrix	Component Factor Loadings							
	1	2	3	4	5	6	7	8
Spatial_orientation_own	0.695							
Spatial_orientation_other	0.751							
2_Objects_spatial_orientation	0.721							
Objects_spatial_orientation	0.721							
Map_spatial_orientation	0.715							
Spatial_orientation_total	0.845							
Spontaneous _time	0.863							
Auditory visual kinesthetic	0.783							
Audio_perceptual_ motors	0.877							
Rhythm_gait	0.877							
Rhythm _total	0.926							
Test beat_quality	0.797							
Test beat_duration	0.797							
Auditive_attention_total	0.797							
Passivity pulse resistance	0.502			0.555				
Passivity on resistance footing	0.482			0.456				
Shoulder strength extensibility	0.406			0.562				
Pulse extensibility endurance	0.544			0.571				
Popliteal extensibility	0.668	0.706						
Adductor extensibility	0.680	0.705						
Heel-ear resistance	0.680	0.705						
ankle contraction	0.685	0.687						
Extensibility foot with bent leg	0.680	0.705						
Extensibility foot with stretched leg	0.680	0.705						
Trunk extensibility	0.680	0.705						
Rotulien_reflexes_right_leg	0.623	.597						
Rotulien_reflexes_left_leg	0.623	0.597						
Passive_mobilization	0.685	0.687						
Residual_tonus _total	0.752	0.627						
Support_ tonus_ total	0.685	0.687						
Attitude_tonus _total	0.469	0.053				0.383		
diadochokinesis pronation_supination	0.685	0.687						
Open_close_hands	0.486	0.428						
Open_close_mouth	0.486	0.428						
Syncinese_3tests	0.685	0.687						
Syncinese	0.685	0.687						
Action_Tonus _total	0.606	0.563						
Tonus_total	0.761	0.635						
Dynamic balance	0.861							
Spontaneous gait	0.279							0.597
Walking to the front in a line	0.768			0.124				
Walking to back in a line	0.531				0.097			
Tiptoe gait	0.815		0.068					
Heels gait	0.861				0.067			
8 inches/20 cm jump	0.720							0.083
Upper and lower limbs coordination	0.720							0.083
Static balance	0.942		0.145					
Immobility	0.886				0.066			
Single-leg support	0.729		0.332					
Tiptoe immobility	0.817		0.291					
Global_Motricity_total	0.915							0.078
Symmetrical bimanual supination pronation	0.843					0.048		
Asymmetrical bimanual supination pronation	0.842		0.061					
Bimanual supination pronation	0.690					0.131		
Forefinger_thumb	0.717					0.107		
Thumb opposition	0.694					0.111		
Manual and digital Praxis_total	0.856					0.081		
Tactile_Gnosis_right_hand	0.822		0.437					
Tactile_Gnosis_left_hand						0.529	0.500	
Tactile_Gnosis_total	0.609		0.419					
Hand_eye performance right hand	0.743						0.190	
Hand_eye performance left hand	0.511						0.409	
Hand-eye Coordination_total	0.666						0.343	
Gestual_Laterality	0.543				0.429			
Laterality_total	0.543			0.429				

**Table 7 children-09-01195-t007:** Means, standard deviation and Student’s t-test of NPmot.pt domains by gender and diagnosis (*n* = 200).

NPmot.pt	Gender Female (*n* = 105)	Gender Male (*n* = 95)		Group with DD (*n* = 50)	Group without DD (*n* = 150)	
	M ± sd	M ± sd	*p*	M ± sd	M ± sd	*p*
Tonus	34.29 ± 5.02	33.78 ± 5.14	0.48	28.92 ± 7.80	35.75 ± 1.63	<0.01
Gross Motor Skills	12.47 ± 3.89	12.0 ± 4.5	0.43	6.24 ± 1.71	14.25 ± 2.51	<0.01
Laterality	4.85 ± 0.04	4.9 ± 0.25	0.04	4.56 ± 0.5	5.00 ± 0.00	<0.01
Manual Praxis	6.06 ± 2.8	6.06 ± 2.33	0.88	3.02 ± 1.00	7.03 ± 2.09	<0.01
Hand-eye coordination	2.91 ± 1.4	2.64 ± 0.82	0.12	1.34 ± 0.71	3.27 ± 0.9	<0.01
Tactile Gnosis	2.80 ± 1.54	2.3 ± 1.55	0.06	0.82 ± 0.92	3.18 ± 1.24	<0.01
Spatial Orientation	2.82 ± 1.63	2.66 ± 1.72	0.51	0.74 ± 0.75	3.41 ± 1.33	<0.01
Rhythm	3.74 ± 1.60	3.19 ± 1.91	0.03	0.54 ± 0.5	4.46 ± 0.50	<0.01
Auditive Attention	3.14 ± 1.46	2.76 ± 2.03	0.22	0.52 ± 0.89	3.77 ± 1.10	<0.01
Total	8.12 ± 2.15	7.81 ± 2.06	0.31	5.19 ± 1.64	8.90 ± 1.26	<0.01

*p* < 0.05; DD = developmental disorders.

**Table 8 children-09-01195-t008:** Mean and standard deviation of NPmot.pt domains by age.

NPmot.pt	4.0–4.9 (*n*_TP_ = 29)	4.10–5.8 (*n* = 41; *n*_TP_ = 36; *n*_DD_ = 5	5.9–6.6 (*n*_TP_ = 7)	6.7–7.5 (*n* = 48 *n*_TP_ = 37 e *n*_DD_ = 11)	7.6–8.5 (*n* = 52 *n*_TP_ = 41 e *n*_DD_ = 11)	>8.5 (*n* = 23 *n*_TP_ = 0 e *n*_DD_ = 23)
	M ± sd	M ± sd	M ± sd	M ± sd	M ± sd	M ± sd
Tonus	35.72 ± 0.82	34.10 ± 5.17	36.00 ± 0.00	34.10 ± 4.99	35.13 ± 2.76	28.65 ± 8.62
Gross Motor Skills	11.48 ± 0.63	11.51 ± 1.27	11.71 ± 0.49	14.08 ± 3.95	14.60 ± 4.70	5.52 ± 1.12
Laterality	5.00 ± 0.00	5.00 ± 0.00	5.00 ± 0.00	5.00 ± 0.00	4.87 ± 0.35	4.35 ± 0.49
Manual and Digital Praxis	4.93 ± 0.25	4.66 ± 0.94	5.43 ± 0.79	7.31 ± 2.62	7.67 ± 3.01	3.65 ± 0.78
Hand-eye coordination	3.07 ± 0.26	2.93 ± 0.41	3.29 ± 0.49	2.94 ± 1.66	3.06 ± 1.04	1.09 ± 0.79
Tactile Gnosis	2.97 ± 0.26	2.93 ± 0.41	3.29 ± 0.49	2.94 ± 1.66	3.06 ± 1.04	1.09 ± 0.79
Spatial Orientation	2.00 ± 0.00	1.95 ± 0.22	3.43 ± 0.98	3.46 ± 1.46	4.04 ± 1.80	0.48 ± 0.67
Rhythm	4.00 ± 0.00	3.54 ± 1.27	4.00 ± 0.00	4.00 ± 1.82	3.83 ± 1.94	0.70 ± 0.47
Auditive Attention	3.79 ± 0.62	2.68 ± 1.46	3.71 ± 0.76	3.25 ± 1.96	3.35 ± 1.76	0.70 ± 0.97
Total	8.11 ± 0.32	7.7 ± 1.11	8.43 ± 0.44	8.56 ± 2.24	8.85 ± 2.04	5.14 ± 1.63

TP = typical development; DD = developmental disorders.

**Table 9 children-09-01195-t009:** Post-hoc Scheffe test scores for the age ranges.

NPmot.pt	G1 vs. 2	G1 vs. 3	G1 vs. 4	G.1 vs. 5	G1 vs. 6	G2 vs. 3	G2 vs. 4	G2 vs. 5	G2 vs. 6	G3 vs. 4	G 3 vs. 5	G3 vs. 6	G 4 vs.5	G4 vs. 6	G5 vs. 6
T	0.84	1	0.83	1	<0.001	0.96	1	0.95	<0.001	0.96	1	0.03	0.94	<0.001	<0.001
GM	1	1	0.04	<0.001	<0.001	1	0.02	<0.001	<0.001	0.64	0.41	<0.001	0.99	<0.001	<0.001
L	1	1	1	0.33	<0.001	1	1	0.22	<0.001	1	0.86	<0.001	0.17	<0.001	<0.001
MDP	1	1	<0.001	<0.001	0.43	0.98	<0.001	<0.001	0.63	0.42	0.21	0.56	1	<0.001	<0.001
TG	1	1	1	1	<0.001	0.98	1	1	<0.001	0.98	1	<0.001	1	<0.001	<0.001
HEC	0.58	0.76	1	1	<0.001	1	0.24	0.45	0.04	0.62	0.75	0.75	1	<0.001	<0.001
SO	1	0.17	<0.001	<0.001	<0.001	0.12	<0.001	<0.001	<0.001	1	0.90	<0.001	<0.001	<0.001	<0.001
R	0.89	1	1	1	<0.001	0.99	0.82	00.97	<0.001	1	1	<0.001	1	<0.001	<0.001
AA	0.12	1	0.81	0.90	<0.001	0.75	0.70	0.51	<0.001	1	1	<0.001	1	<0.001	<0.001
Total	0.83	0.88	0.63	0.58	0.049	0.86	0.53	0.46	0.08	0.85	0.79	0.15	0.79	<0.001	<0.001

*p* < 0.05; Subtitle: Group 1 = 4.0–4.9 years; Group 2 = 4.10–5.8 years; Group 3 = 5.9–6.6 years; Group 4 = 6.7–7.5 years; Group 5 = 7.6–8.4 years; Group 6 ≥ 8.5 years; T = Tonus; GM = Gross Motor Skills; L = Laterality; MDP = Manual and Digital Praxis; TG = Tactile Gnosis; HEC = Hand-eye coordination; SO = Spatial Orientation; R = Rhythm; AA = Auditive Attention.

## Data Availability

The datasets used and analyzed in this study are available from the corresponding author on reasonable request.

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
