# Peer review of "The Assessment of the Psychomotor Profile in Children: Preliminary Psychometric Analysis of the Portuguese Version of the Batterie d’Evaluation des Fonctions Neuropsychomotrices de L’enfant (NPmot.pt)"

_children, 2022, doi:10.3390/children9081195_

Round 1

Reviewer 1 Report

It is a very complete and well-done manuscript, it includes evidence and reliability and validity, the latter from three perspectives. Ethical precepts have been properly followed. The methodologies used are correct and the results are based on them. The discussion and conclusions include limitations, future studies and significant contributions. I would only have one recommendation: most of the statistics used (for example ANOVA, student's t or maximum likelihood factor analysis) are based on assumptions such as normality, homoscedasticity or linearity, the authors do not make any reference to the fulfillment of these assumptions I recommend including them.

Author Response

Dear Editor,

 Title: The assessment of the psychomotor profile in children: preliminary psychometric analysis of the Portuguese version of the Battery of child neuropsychomotor function evaluation BPmot

We appreciate the opportunity to resubmit our article. All comments by reviewers have been addressed, with corresponding changes made directly to the manuscript where appropriate. We think that these recommendations are very important and have substantially strengthened the article. Accompanying this letter, please find a revised version of our manuscript. Please, also see below for a response to the Editor’s and Reviewer’s comments.

Yours sincerely,

The authors

Response to Editor’s Comments

We would like to thanks the reviewers for all comments and recommendations, which were  very important and have substantially strengthened the article. After reading very carefully all comments, critiques and recommendations we are able to say that the changes suggested by the reviewers were made: in terms of content as well as format.

Reviewer 1 – Major Revisions

  1. It is a very complete and well-done manuscript, it includes evidence and reliability and validity, the latter from three perspectives. Ethical precepts have been properly followed. The methodologies used are correct and the results are based on them. The discussion and conclusions include limitations, future studies and significant contributions.

Answer/comment: thank you for the remark.

  1. I would only have one recommendation: most of the statistics used (for example ANOVA, student's t or maximum likelihood factor analysis) are based on assumptions such as normality, homoscedasticity or linearity, the authors do not make any reference to the fulfillment of these assumptions I recommend including them.

Answer: noted and corrected; in the text were added a simple explanation about all these questions raised by the reviewer:

The choice of tests for the statistical analysis of the data collected was based on the fulfillment of the assumptions necessary for parametric tests. In this sense, the dependent variables presented a normal distribution in both groups, most of the cases being close to the mean value. The existence of homogeneity of variances (homoscedasticity) was also verified, that is, the variances of the dependent variables were homogeneous in all groups. The test used to verify Normality was the Kolmogorov-Smirnov, and the Levene test was used to test the homogeneity of variances. The sample size (n=200) also consolidated the decision to use parametric tests.

Reviewer 2 Report

Dear authors,

first of all thanks for the submission made to Children.

The authors performed a preliminary analysis of the psychometric properties of the Portuguese version of the Battery for Neuropsychomotor functions evaluation, and deserve credit for the way the work is structured.

However, I believe that there are some points that can be improved in order to increase the potential of the work.

1) keywords:

Try to reformulate the first keyword since it is already in the title

2) Introduction:

I believe the introduction is too long. Although it is well organized from a structural point of view, I believe it would benefit from being more assertive. This is an experimental article and not a review article for these reasons it would be more beneficial if the framing was carried out objectively, I believe that this can facilitate the understanding of the study problem for the reader.

3) Materials and Methods

This point is very well described.

4)Results:

Perhaps the information placed in the first 2 paragraphs of the results should incorporate the "Material and methods" tab. in fact I believe that at this point only the results of the study should be exposed.

Attention with some formatting problems that are denoted in the tables. Also, try to be consistent by keeping the same type of formation on all tables.

5) Discussion and conclusion

The discussion and conclusion are well structured and justify the study problem, although perhaps the limitations of the study could appear at the end of the discussion and not in the conclusion. Check the formation at the end of the discussion.

Author Response

Dear Editor,

 Title: The assessment of the psychomotor profile in children: preliminary psychometric analysis of the Portuguese version of the Battery of child neuropsychomotor function evaluation BPmot

We appreciate the opportunity to resubmit our article. All comments by reviewers have been addressed, with corresponding changes made directly to the manuscript where appropriate. We think that these recommendations are very important and have substantially strengthened the article. Accompanying this letter, please find a revised version of our manuscript. Please, also see below for a response to the Editor’s and Reviewer’s comments.

Yours sincerely,

The authors

Reviewer 2

The authors performed a preliminary analysis of the psychometric properties of the Portuguese version of the Battery for Neuropsychomotor functions evaluation, and deserve credit for the way the work is structured. However, I believe that there are some points that can be improved in order to increase the potential of the work.

Answer/comment: thank you for the remark.

  • Keywords: Try to reformulate the first keyword since it is already in the title

Answer: according to reviewer suggestion the first keyword was deleted and added: Evaluation; Psychomotor Development;

  • Introduction: I believe the introduction is too long. Although it is well organized from a structural point of view, I believe it would benefit from being more assertive. This is an experimental article and not a review article for these reasons it would be more beneficial if the framing was carried out objectively, I believe that this can facilitate the understanding of the study problem for the reader.

Answer/comment: indeed we were concerned with a complete “picture” of the psychomotor development; according to reviewer comments the introduction were shortened; all changes and adaptations to the text are presented in the text with track changes.

  • Materials and Methods: This point is very well described.

Answer/comment: thank you for the remark.

  • Results: Perhaps the information placed in the first 2 paragraphs of the results should incorporate the "Material and methods" tab. in fact I believe that at this point only the results of the study should be exposed.

Answer/comment: done; both paragraphs mentioned by the reviser were anticipated and written in Material and Methods.

  • Attention with some formatting problems that are denoted in the tables. Also, try to be consistent by keeping the same type of formation on all tables.

Answer/comment: done; was given a special attention to the format of tables

  • Discussion and conclusion - The discussion and conclusion are well structured and justify the study problem, although perhaps the limitations of the study could appear at the end of the discussion and not in the conclusion. Check the formation at the end of the discussion.

Answer: noted and corrected.  The paragraph mentioned by the reviser were anticipated and written at the end of Discussion.

Round 2

Reviewer 2 Report

Dear authors I belive that you have done a good job. I'm this sense I have no further recommendations. 

Author Response

Dear Editor,

 Title: The assessment of the psychomotor profile in children: preliminary psychometric analysis of the Portuguese version of the Battery of child neuropsychomotor function evaluation BPmot

We appreciate the opportunity to resubmit our article. All comments have been addressed, with corresponding changes made directly to the manuscript where appropriate. We think that these recommendations are very important and have substantially strengthened the article. Accompanying this letter, please find a revised version of our manuscript. Please, also see below for a response to the Editor’s comments.

Yours sincerely,

The authors

Response to Editor’s Comments

We would like to thanks the editor for all comments and recommendations, which were  very important and have substantially strengthened the article. After reading very carefully all comments, critiques and recommendations we are able to say that the changes suggested by the editor were made: in terms of content as well as format.

Editor – Minor Revisions

Introduction

  1. The introduction is too long for a validation paper. The authors should be more concise when presenting their ideas. The main focus should be the NP-MOT. Accordingly, the reason for having a neuropsychomotor assessment instrument, instead of a psychomotor assessment instrument, should be explained.

Answer: noted and corrected; in the text were added a simple explanation about these questions raised by the editor. According to editor comments the introduction were shortened; all changes and adaptations to the text are presented in the text with track changes.

“Currently, there is a large group of researchers [4–8] who believe that Psychomotricity should follow the evolution of neuroscience. Human behavior is a reflection of its internal organization and relies on neurological substrates that contribute to a better understanding of human development. For this reason we believe in this new conceptual framework, and will use the concept neuropshychmotor throughout this article.”

This lack of clarity, along with the inconsistent use of terms (psychomotor/neuropsychomotor; see for example paragraph staring in L171) might be confusing to the reader.

Answer: noted and corrected; in the text, the term “psychomotor” was changed to “neuropsychomotor”.

  1. Throughout the manuscript, the authors use similar terms: psychomotor limitations, psychomotor developmental problems, psychomotor symptoms. Are they referring to the same concept? A higher consistency in the terms would improve the clarity of the manuscript. Answer: noted and corrected; in the text, the term “psychomotor limitations/ psychomotor developmental problems/ psychomotor symptoms” was changed to “psychomotor limitations”.

 It was changed for a better reading of the article, however, all of these words are synonymous.

  1. I believe there is a critical conceptual statement in the introduction. In the first paragraph, the authors state that psychomotricity has been renamed to neuropsychomotrity. This is an inadequate statement that can have a negative effect on the social acceptance of such a recent area. This paragraph should be carefully reformulated.

Answer: noted and corrected; this sentence was deleted from the article.

  1. Also in this introductory paragraph, psychomotricity is presented as uniquely focused on the child (lines 44, 45). This should be adjusted, as psychomotricity is focused on the entire lifespan.

Answer: noted and corrected; the word “child” was changed to “person”.

  1. The statement “That results from…” (line 44) is incomplete and not clear.

Answer: noted and corrected, the sentence was changed to clarify the meaning of that.

“PMT, integrating cognition, body movement, and emotions, is based on learning-by-movement experiences in a holistic view of the child, through the unity of body and mind.”

  1. Also, the last statement seems repetitive and only presents a functional perspective of psychomotricity : “is focused on psychomotor skills that imply the use of psychomotor skills in daily activities”. PM is also focused on the experience of the body with no functional purpose. I suggest the authors adjust this sentence.

Answer: noted and corrected; this sentence was deleted from the article.

  1. In line 188 the authors refer “psychic processes”. Which processes are they referring to? This term does not seem aligned with the theoretical framework underlying the NP-MOT.

Answer: noted and corrected; the word “psychic processes” was changed to “thought”.

  1. The paragraph starting on line 192 could be summarized and integrated with the following paragraph.

Answer: noted and corrected.

  1. For example, for reasons of clarity, I believe that the statements regarding the Portuguese psychomotor battery can be removed (lines 196-199) as they are not specifically related to the NP-MOT.

Answer: noted. However, we think that it is important to keep this statement since it contextualizes the assessment process in Portugal.

Methods

  1. How was the sample size chosen in order to answer the research question? Was a power analysis conducted (not per se required, but there should be a rationale for the chosen sample size)? If not, this should be recognized as a limitation of the study.

Answer: noted and corrected. The sentence was changed, and the present situation is considered now a study’s limitation.

  1. Were outliers removed from the data?

Answer: noted, and yes. There are no outliers.

  1. Provide information about missing data and how missing cases are dealt with in the analyses. Make sure the reported n is of the actual analyses and indicate whether (and, if so, why) it differs from the sample size as reported in the methods section.

Answer: noted, and there are no missing data.

  1. Besides Means and SDs, also provide min-max scores in the participants’ section, regarding each group (TD and DD).

Answer: thank you for the remark, but we do not consider this information relevant for this article.

  1. The authors used two samples: children with typical development and children with developmental disorders. Therefore, the inclusion/exclusion criteria should be presented for both samples. For example, were children with ADHD included? In which sample? Also, the authors should give some examples of the developmental disorders included in the DD sample.

Answer: noted, but all the inclusion and exclusion criteria were presented in the correct section.

  1. Test-retest was performed with children with typical development? How were these children selected? This should be clarified. Also, this information should be uniquely presented in the procedures and therefore removed from the participants’ section.

Answer: noted. To realize this technique, it was used 50 children from the total sample. The information about their characterization (with or without DD, wasn’t relevant). The goal was the same, to analyze the reliability of the test.

  1. The first paragraph regarding the NP-MOT is a theoretical presentation of the instrument and should appear in the introduction, not in the methods.

Answer: noted and corrected. The sentence was changed to the Introduction section.

  1. In line 312 the authors state that the Portuguese version of the NP-MOT “maintained the same original structure”. Considering the aims of the study, the authors should present this information in the results section.

Answer: thank you for the remark, but the purpose of this sentence was to state that in the study we assume the same structure as the original test.

  1. The last paragraph of the “Instrument” section should be summarized. If the reader needs such a detailed description of the original instrument, he/she can read the original validation paper.

Answer: thank you for the remark, but these values help the discussion and so it was decided to keep them.

  1. The reference of the Lincoln-Bruininsky-Oseretsky Test of Motor Proficiency Battery (Line 340) is missing.

Answer: noted and corrected.

  1. Regarding the ethical procedures (line 374) the authors state that participants signed the consent form. Considering the age and developmental characteristics of the participants, did the examiners also ask for children’s written oral consent? Were these procedures adapted to the children with developmental disorders? These details are important and should be added.

Answer: noted and corrected.

  1. Accordingly, in line 377 the authors present an average time of application. Is this referring to the children with typical development? Or did they average the application time for all participants? It would be important to specify the time of application for both samples, and perhaps, a minimum and maximum time.

Answer: noted. This average time of application refers to all the participants.

  1. Also, were there adaptations to the application of the NP-MOT to the children with developmental disorders? This information is important.

Answer: noted. There were no adaptations for children with DD. All items were explained and applied in the same way to all participants.

  1. In line 417 the authors state that “The application of the battery (…) not obeying any specific strict order”. How was the battery applied? Did the examiners follow the same order? This should be specified in the procedures. Also, are there necessary conditions (area, instruments, free wall, etc.,) for the assessment room? I could not find this information which I find critical for PM therapists and researchers who are interested in this instrument.

Answer: noted. In this specific test, there is no standard protocol to apply. All domains can be applied independently. The examiners applied all items. The room conditions are the basic conditions, to perform a neuropsychomotor assessment: a bright room, with an adequate temperature and with all the necessary material (formal and informal).

Results

  1. The titles of the tables should be more informative, and present the notation of the statistics and the sample size. In the title should be also clear if the statistical analyses were performed to the whole sample or to a specific sample.

Answer: noted and corrected.

  1. The title of Table 3 is written in Portuguese and should be adjusted.

Answer: noted and corrected.

  1. Table 9 should not have values in bold.

Answer: noted and corrected.

  1. The names of the domains should be revised as they appear with different names (e.g., Bi-manual and digital praxis in Table 4 and manual and digital praxis in Table 5).

Answer: noted and corrected.

  1. The authors refer to a 8-domains structure, contrary to the French 9-domains structure. More details should be presented referring to the domains that did not fit the original structure.

Answer: thank for the remark, however, the domain in question, in all the analyses performed, proved not to be very robust or relevant. Possible reasons for this are that the domain is represented by only one item and/or that it may incorporate a broader domain. This situation seems to justify the possible removal of one of the 9 domains from the test framework.

  1. The authors carefully present the differences between children with typical development and children with developmental disorders regarding the psychomotor profile. Actually, they test discriminant validity supported on these differences. Therefore, presenting the descriptive for age groups considering the whole group of children is contradictory. Indeed, the authors seem to assume this problem later, when justifying the “Lower values (…) in the older age group” which have “a high percentage of children with developmental disorders”. I believe these analyses should be performed separately for each group (TD and DD).

Answer: Thank you for your comment, however no analysis by subgroup was performed, since the n's of the sample subgroups were too divergent, not allowing objective conclusions to be drawn. For this reason, we kept only the two subgroups: with and without developmental disorders.

Discussion

  1. The discussion should be shortened and focused on the findings of the present study.

Answer: noted and corrected.

  1. The 1st paragraph of the discussion is very interesting and appealing. However, I believe it should appear in the end of the discussion or at the end of the introduction as it focuses the potential of the study. It would be worthwhile if the authors could summarize the findings of their study in the 1st paragraph.

Answer: noted and corrected.

  1. Accordingly, in the 2nd paragraph the authors present the theoretical and methodological support of their study. Although important, this information is repetitive and should be integrated into the previous sections.

Answer: noted and corrected.

Conclusion

  1. The 1st paragraph of the conclusion goes to the point in a very effective and clear way! However, such a clear and focused approach got lost in the 2nd paragraph where previous ideas are repeated. I suggest the authors reformulate this last part, in line with the 1st paragraph.

Answer: noted and corrected. The paragraph was simplified.

  1. Tables Make sure that the following information is available in all relevant table legends: ∙ the exact sample size (n) for each group; ∙ definitions of statistical symbols; ◦ statistical test results, e.g. P values;

Answer: noted and corrected.

  1. Finally, I have a concern regarding the authorship of this manuscript, as the second author does not appear in the contributions. Everyone who is listed as an author should have made a substantial, direct, intellectual contribution to the work.

Answer: noted and corrected.
